# Characterization of water-soluble brown carbon chromophores from wildfire plumes in the western US using size exclusion chromatography

Lisa Azzarello[1], Rebecca A. Washenfelder[2], Michael A. Robinson[2,3], Alessandro Franchin[2,3,4], Caroline C. Womack[2,3], Christopher D. Holmes[5] Steven S. Brown[2,6], Ann Middlebrook[2], Tim Newberger [7], Colm Sweeney [7], Cora J. Young[1]

[1]Department of Chemistry, York University, Toronto, ON, M3J 1P3, Canada

[2]Chemical Sciences Laboratory, National Oceanic and Atmospheric Administration, 325 Broadway, Boulder, CO 80305, USA

[3]Cooperative Institute for Research in Environmental Sciences (CIRES), University of Colorado, Boulder, CO, 80309, USA

[4]Now at: National Center for Atmospheric Research, Boulder, CO, USA

[5]Earth, Ocean, and Atmospheric Science, Florida State University, Tallahassee, FL 32304, USA

[6]Department of Chemistry, University of Colorado Boulder, Boulder, Colorado, USA

[7]Global Monitoring Laboratory, National Oceanic and Atmospheric Administration, 325 Broadway, Boulder, CO 80305, USA

*Correspondence to:* C. J. Young (youngcj@yorku.ca)

**Abstract**

Wildfires are an important source of carbonaceous aerosol in the atmosphere. Organic aerosol that absorbs light in the ultraviolet to visible spectral range is referred to as "brown carbon" (BrC), and its impact on Earth's radiative budget has not been well characterized. We collected water-soluble brown carbon using a particle into liquid sampler (PILS) onboard a Twin Otter aircraft during the Fire Influence on Regional to Global Environments and Air Quality (FIREX-AQ) campaign. Samples were collected downwind of wildfires in the western United States from August to September 2019. We applied size exclusion chromatography (SEC) with ultraviolet-visible spectroscopy to characterize the molecular size distribution of BrC chromophores. The wildfire plumes had transport ages of 0 to 5 h and the absorption was dominated by chromophores with molecular weights <500 Da. With BrC normalized to a conserved biomass burning tracer, carbon monoxide, a consistent decrease in BrC absorption with plume age was not observed during FIREX-AQ. These findings are consistent with the variable trends in BrC absorption with plume age reported in recent studies. While BrC absorption trends were broadly consistent between the offline SEC analysis and the online PILS measurements, the absolute values of absorption and their spectral dependence differed. We investigate plausible explanations for the discrepancies observed between the online and offline analyses. This included solvent effects, pH, and sample storage. We suspect that sample storage impacted the absorption intensity of the offline measurements without impacting the molecular weight distribution of BrC chromophores.

## 1. Introduction

The wildfire season across the western United States has greatly intensified over the past century. The U.S. Forest Service reports that the amount of western U.S. land burned by "high severity" wildfires (i.e., fires that destroy more than 95% of vegetation) has increased eightfold since 1985 (Parks and Abatzoglou, 2020). A variety of factors influence the number and intensity of wildfires, including fuel availability, temperature, drought conditions, location of lightning strikes, and direct human influence. During the 20[th] century, fire suppression tactics were applied throughout the western U.S. and these efforts caused fuel to accumulate (Marlon et al., 2012). The combination of dry conditions, warmer temperatures, and fuel availability contributes to the intensity of present-day wildfires in the western U.S. Consequently, the impact that these climatic conditions have on wildfire activity has been established. However, feedback effects that wildfires have on climate is an ongoing area of research.

Wildfires emit carbonaceous particulate matter into the atmosphere (Bond et al., 2004; van der Werf et al., 2010). Based on volatility and optical properties, carbonaceous aerosol particles emitted from biomass burning are categorized as elemental carbon (EC) and organic carbon (OC) (Turpin et al., 1990). Elemental carbon, referred to as black carbon (BC), is refractory and is characterized by broad absorbance across the ultraviolet (UV) to infrared wavelengths (Seinfeld and Pankow, 2003; Andreae and Gelencsér, 2006; Laskin et al., 2015). The light-absorbing components of organic aerosols are referred to as brown carbon (BrC) (Laskin et al., 2015). The direct absorption and scattering of solar radiation by these aerosol particles impacts the global radiative budget (Boucher, O.; Randall, D.; Artaxo, P.; Bretherton, C.; Feingold et al., 2013; Forster, P.; Ramaswamy, V.; Artaxo, P.; Berntsen, T.; Betts et al., 2007), but there is uncertainty about the magnitude of this effect. Currently, more information is known about BC and its impact on climate than BrC, as BrC is more chemically complex and more reactive (Buis, 2021; Di Lorenzo et al., 2017). The direct radiative forcing of BC has been estimated to be the second largest anthropogenic climate forcing species (Ramanathan and Carmichael, 2008) and studies have suggested that BrC can contribute between 20 to 40 % to positive radiative forcing from total carbonaceous absorbing aerosol (Feng et al., 2013; Zhang et al., 2017; Zeng et al., 2020a).

Wildfire emissions are a dominant primary source of BrC (Washenfelder et al., 2015). The brown colour results from a combination of species with varying abilities to absorb light in the UV-visible region (from highly to weakly absorbing) (Hems et al., 2021). The pyrolysis of lignin

and cellulose contributes to the major chemical constituents in wildfire plumes, such as phenolic
compounds and organic acids (Simoneit, 2002; Xie et al., 2019; Smith et al., 2014). Lignin
pyrolysis products with aromatic functionalities absorb visible light and may contribute to the
absorption properties of BrC (Hems et al., 2021). Secondary processes also contribute to BrC
formation. The generation of secondary organic aerosol (SOA) stemming from gas phase reaction
products includes nitration of aromatic compounds in the presence of $NO_x$ or $NO_3$ (Harrison et al.,
2005; Finewax et al., 2018; Xie et al., 2017). For example, catechol can react with either the $NO_3$
or OH radical to form 4-nitrocatechol (Finewax et al., 2018) and oxidation of toluene under
elevated $NO_x$ conditions has been observed to form over 15 absorbing compounds with
nitroaromatics contributing up to 60% of absorption in the visible region (Liu et al., 2016).
Although there are hypotheses about the identity of BrC chromophores, up to 40% of BrC
constituents remain unidentified (Lin et al., 2017; Bluvshtein et al., 2017).
To characterize the absorbing constituents that contribute to BrC absorption, reverse phase
high performance liquid chromatography (HPLC) coupled to high resolution mass spectrometry
has been applied (Fleming et al., 2020). However, fresh and aged BrC consist of extremely low
volatile organic compounds (ELVOCs) that may be irreversibly retained on a traditional $C_{18}$
reverse phase HPLC column (Di Lorenzo and Young, 2016). Size exclusion chromatography
coupled to ultraviolet-visible absorption spectroscopy (SEC-UV) has been demonstrated as an
alternative that successfully measures the absorption properties of high and low molecular weight
(MW) ELVOCs in fresh and aged biomass burning-derived samples (Di Lorenzo and Young,
2016; Di Lorenzo et al., 2017; Wong et al., 2019). Analysis by SEC-UV has been previously
applied to samples collected during ground-based field measurement campaigns, where the
method has established the quantification of BrC absorbance as a function of MW and provided
insight into the composition of BrC. High MW (>400 Da) compounds with unknown structural
identities have been determined to contribute to BrC absorption and the relative contribution to
BrC absorption by high MW species increases with smoke age (Di Lorenzo et al., 2017; Wong et
al., 2017, 2019). These findings suggested that lower MW species are less persistent in biomass
burning smoke relative to higher MW species, likely due to volatilization, oxidation,
polymerization, or other processes (Di Lorenzo et al., 2017; Hems et al., 2021).
The Fire Influence on Regional to Global Environments and Air Quality (FIREX-AQ) field
campaign examined the impact of wildfires on atmospheric chemistry and air quality in the western
United States. In this work, we present the SEC-UV analysis of water-soluble BrC that was
collected on board the National Oceanic and Atmospheric Administration (NOAA) Twin Otter
aircraft during plume transects downwind from western U.S forest fires. These represent the first
aircraft samples analyzed by SEC-UV to characterize BrC. We compare the total absorption
measured in online and offline samples and assign the BrC absorption to different MW classes.
Finally, we examine how the composition of the mobile phase used in the SEC-UV analysis
impacts elution time and spectral features. This provides cautionary information about interpreting
absorption results in studies that apply chromatographic separation in an aqueous-organic matrix.

**2.   Experimental Approach**
**2.1 Overview of the FIREX-AQ field campaign**

FIREX-AQ was a multi-platform field campaign that investigated wildfire emissions in the

western United States from Jul to Sep 2019. Instrumented aircraft and mobile laboratories were
used to intercept and sample smoke plumes throughout multiple western U.S. states. These
included a DC-8, ER-2, and two Twin Otter aircraft. This study focuses on smoke sampled by the
"Chemistry" Twin Otter aircraft, which was based in Boise, Idaho, from 29 Jul to 5 Sep 2019, and
briefly in Cedar City, Utah, from 19 Aug to 23 Aug 2019. The Twin Otter payload included gas
and aerosol instruments to measure smoke composition, transport, and transformation. This
included CO measurements by near infrared cavity ring-down spectroscopy (G2401-m; Picarro
Inc., Santa Clara, CA, USA) (Crosson, 2008; Karion et al., 2013). A complete description of the
payload installed on the Twin Otter can be found in Warneke et al. (2023). The payload weight
limited the duration of in-flight sampling to $2.5 - 3$ h, with a typical schedule of two or three flights
per day during the afternoon, evening, or night. A total of 40 flights were completed in Arizona,
Idaho, Nevada, Oregon, and Utah. Airmass back trajectory analyses were used to estimate the
plume age of each transect, as described in Liao et al. (2021) and Washenfelder et al. (2022).
Briefly, the smoke age was calculated by summing the horizontal advection and vertical plume
rise ages between the time of emission and the aircraft interception of the smoke plume. For the
Twin Otter flights, many plume intercepts by the aircraft were approximately Lagrangian
(Washenfelder et al., 2022).

## 2.2 Online measurement of water-soluble absorption and offline sample collection

The Brown Carbon-Particle into Liquid Sampler (BrC-PILS) collected online absorption data and offline aqueous samples for the SEC-UV analysis. A complete description of the BrC-PILS instrument and sampling can be found in Zeng et al. (2021) and Washenfelder et al. (2022). Briefly, the BrC-PILS sampled smoke through a shared aerosol inlet on the Twin Otter. A parallel-plate carbon filter denuder removed volatile organic compounds prior to the aerosol entering the PILS. The PILS consisted of a steam generator and droplet impactor to collect aerosols into aqueous solution. The liquid flow then entered a liquid waveguide capillary cell (LWCC) to measure absorption. The instrument precision ($3\sigma$) for absorption at 365 nm was $\pm 0.02$ Mm$^{-1}$ for 10 s in-flight data, with an uncertainty of $\pm 11$ % (Zeng et al., 2021). The flow exiting the LWCC was split between a total organic carbon (TOC) analyzer and an automated 14-port valve. The valve directed aqueous sample flow to one of 12 polypropylene sample tubes for offline SEC-UV analysis (Figure S1). Prior to deployment, each polypropylene tube was rinsed with 18.2 M$\Omega \cdot$cm deionized water (DIW) (Thermo Scientific Barnstead Smart2Pure) eight to ten times. The sample flow rate was monitored by a liquid mass flow meter prior to the flow diverting between the automated valve and the TOC analyzer. The sample flow was 1.53 mL min$^{-1}$ during inflight sampling, and the excess 0.43 mL min$^{-1}$ was collected into an individual polypropylene tube for 12 s to 10 min. During in-flight sampling, the flight scientist actively controlled the sample collection into each polypropylene tube to target transects of the smoke plume (example shown in Figure S2). Six to twelve aqueous samples were collected for each flight, with 201 total samples from 39 science flights. Field blanks of the DIW used to operate the BrC-PILS were stored similarly in clean polypropylene sample tubes at the beginning, halfway point, and end of campaign. Once collected in the field, the samples and blanks were stored on ice for several hours prior to refrigeration until analysis.

## 2.3 Offline Analysis by SEC-UV

Measurement by SEC-UV provides information about size-dependent light absorption properties of BrC chromophores. The offline aerosol samples were separated using an aqueous gel-filtration column with a MW range of 250 Da to 75 kDa (Polysep GFC P-3000, Phenomenex, Torrence, CA). Size-resolved components were detected using a diode array detector from 190 to 800 nm (UltiMate 3000, Thermo Scientific, Sunnyvale, CA) coupled to an ion chromatograph

(ICS 6000; Thermo Scientific) pump with an AS autosampler (Thermo Scientific). The isocratic
method was run using a mobile phase that contained a 1:1 mixture of acetonitrile and 25 mM
ammonium acetate in solution at a flow rate of 1 mL min$^{-1}$ and a sample injection volume of 100
µL. A solution of Suwannee River Humic Acid (SRHA II, International Humic Substances
Society, Saint Paul, MN, USA) was run prior to the FIREX-AQ samples to ensure proper operation
of the SEC-UV set-up.

The aqueous samples collected by the BrC-PILS did not require post-sampling processing

and were injected onto the SEC column under mobile phase flow to the diode array detector. The
uncertainty for the offline total absorption measurements considers the uncertainty of the liquid
flow and PILS collection efficiency, for a total uncertainty of ±10.5 %. Discussion of the SEC-UV
method development and details of the conversion of SEC-UV signal to ambient absorption in
units of Mm$^{-1}$ can be found in the SI. We calculated BrC absorption as a function of MW by
applying the calibration method described by Di Lorenzo and Young (2016) (Figure S3). Sample
measurements were blank subtracted. The detection limit of the total integrated absorption
(equivalent to 3σ of n=6 field blanks) was 2.5±0.2 mAU×min and 0.70±0.02 mAU×min at 250 nm
and 300 nm, respectively. This corresponds to a 3σ detection limit of approximately 525 Mm$^{-1}$ at
250 nm and 150 Mm$^{-1}$ at 300 nm.
**2.4 Absorption in different mobile phases**

To assess the impact of pH and mobile-phase composition on wavelength-dependent

absorption, the ammonium acetate solution was adjusted to pH 5 and pH 9 with acetic acid and
ammonium hydroxide, respectively, prior to combining with acetonitrile. A 15 µg/mL in DIW
solution of Suwannee River Fulvic Acid (SRFA II; International Humic Substances Society, Saint
Paul, MN, USA) and a FIREX-AQ aqueous sample were injected onto the diode array detector
without the SEC column in line with the following mobile phases: DIW only; 25 mM ammonium
acetate solution; the default mobile phase (described in Sect. 2.3); 25 mM ammonium acetate
solution adjusted to pH 5; and 25 mM ammonium acetate solution adjusted to pH 9. Solutions of
4-nitrocatechol, 4-hydroxy-3-methoxy cinnamaldehyde, vanillin, and 7-hydroxycoumarin in DIW
with concentrations of $3.9\times10^{-8}$, $3.4\times10^{-8}$, $3.9\times10^{-8}$, $3.7\times10^{-8}$ mol/mL, respectively, were prepared
and injected onto the diode array detector to observe differences in their absorption profiles. To
confirm the diode array detector results, measurements of the SRFA solution were also made with
UV-visible spectroscopy (8453; Agilent Technologies, Santa Clara, CA, USA) where the solution
was mixed (1:1 ratio) with the various mobile phases prior to transferring to a cuvette for
absorption measurements (Figure S7).

## 3. Results and discussion

### 3.1. Trends in absorption with plume age

We present molecular size-resolved absorption for flights that met the following criteria:

(1) maximum CO concentrations greater than 0.2 ppmv; (2) three or more downwind plume
transects; (3) three or more aqueous samples collected; and (4) consistent wind direction. Of the
201 aqueous samples collected, 47 samples from six flights met the criteria and are summarized
in Table S1. Each aqueous sample had a measurable absorption signal in the deep UV region (250
to 300 nm), while the absorption signal above 300 nm was nearly indistinguishable from the
blanks, because the samples were relatively dilute. The average (±standard deviation) integrated
absorption of the 47 samples that met the criteria was 10.4±4.9 mAU×min (8134±3857 $Mm^{-1}$) and
0.36±0.28 mAU×min (316±214 $Mm^{-1}$) for 250 and 300 nm, respectively.

To account for plume dilution, we follow the convention of normalizing BrC absorption to

a conserved tracer, to calculate $\Delta Abs_{\lambda,BrC}/\Delta CO$ (Forrister et al., 2015; Di Lorenzo et al., 2017;
Washenfelder et al., 2022; Zeng et al., 2021; Sullivan et al., 2022), where $\Delta CO$ is the average CO
mixing ratio measured during each aqueous sample collection subtracted by the average CO
background outside the plume. Background BrC absorption at 365 nm (a common wavelength to
report BrC absorption) was less than 0.2 $Mm^{-1}$ and no background correction was made to
determine $\Delta Abs_{\lambda,BrC}$ (Washenfelder et al., 2022). The average CO and variation of CO measured
for each flight are shown in Figure S9. Figure 1 shows $\Delta Abs_{300nm,BrC}/\Delta CO$ as a function of plume
age for the six selected flights, with a linear fit to each flight. The fitted slopes for $\Delta Abs_{300nm,}$
$_{BrC}/\Delta CO$ vs plume age vary from -0.21 to 0.88 $Mm^{-1}$ $ppbv^{-1}$ $h^{-1}$, and show different trends between
flights. This indicates that BrC absorption increased downwind in some plumes and decreased
downwind in others.

Previous studies of normalized BrC absorption with plume age have reported conflicting

results. In the earliest aircraft study, Forrister et al. (2015) collected filter samples from two fires
in the western U.S. and measured the BrC absorption from water and methanol extracts. They
observed that BrC absorption at 365 nm decayed exponentially over a plume age range spanning
0 to 50 h (Figure S10) (Forrister et al., 2015). Di Lorenzo et al. (2017) reported total absorption of
size-resolved BrC chromophores using SEC-UV from three locations that were influenced to
varying degrees by biomass burning smoke, and observed minimal $\Delta Abs_{\lambda,BrC}/\Delta CO$ change as a
function of transport times from 10 to >72 h (Figure S10). In contrast to these measurements of
relatively aged biomass burning aerosol, two studies from other FIREX-AQ instruments showed
different trends for relatively fresh plumes. Using BrC-PILS measurements from the Twin Otter,
Washenfelder et al. (2022) showed variable trends in $\Delta Abs_{365nm,BrC}/\Delta CO$ slope values ranging
from -0.02 to 0.02 Mm$^{-1}$ ppbv$^{-1}$ h$^{-1}$ over 0 to 5 h. Using filter samples from the DC-8 aircraft, (Zeng
et al., 2022) showed that BrC increased, decreased, or was unchanged as a function of plume age
over 0 to 8 h. In another study of fresh plumes, aircraft based measurements during the Western
Wildfire Experiment for Cloud Chemistry, Aerosol Absorption and Nitrogen (WE-CAN; Sullivan
et al., 2022) investigated the evolution of water-soluble BrC at 405 nm normalized to CO and
observed BrC depletion with a smoke age of <2 h, and PILS water-soluble BrC absorption that
broadly remained stable for a smoke age up to 9 h (Sullivan et al., 2022).
Our results are broadly consistent with measurements from FIREX-AQ and WE-CAN that
sampled fresh plumes, and differ from the studies that sampled more aged smoke. The relatively
limited plume age range of the FIREX-AQ sampling makes it challenging to deduce long-term
trends associated with changes in total absorption as a function of transport time. In addition, the
disparity in $\Delta Abs_{\lambda,BrC}/\Delta CO$ time dependence between FIREX-AQ observations and those reported
by Forrister et al. (2015) may be attributed to i) FIREX-AQ having sampled a greater number of
western U.S forest fires; and ii) the younger age of the FIREX-AQ plumes. More in-flight sampling
would be required to observe BrC absorption of plume ages 5 h to 50 h to determine if the results
observed by Forrister et al. (2015) would also show variability with a greater number of fires, or
if the BrC lifetime would converge to a similar value.
**3.2 Chemical evolution of brown carbon with plume age**
Chromophores <500 Da were responsible for most of the absorption at 250-300 nm
measured in the aqueous samples (Figure 2, Figure S11). For the 47 samples, molecular species
>500 Da contributed an average of 3.0±1.9 % to total measured absorption at 250 nm, while
molecular species <500 Da contributed an average of 72±4.5 %. Absorption past the exclusion
volume represents an unidentified MW, as elution past this retention time (Figure S3) indicates
non-SEC analyte-column interactions were occurring. The average contribution to total measured
absorption by undefined MW species was 25.1±5.7 %. Previous SEC-UV analyses have observed
elution beyond the exclusion volume and non-size exclusion effects (Wong et al., 2017; Lyu et al.,

2021). Elution at later retention times has also been observed for fresh BrC separated in a mobile phase containing 50% acetonitrile (Lyu et al., 2021). This result was attributed to non-size exclusion effects, such as hydrophobic interactions of BrC with the SEC stationary phase, which may also have contributed to elution past the exclusion volume in our samples. The absorption density plots of the aqueous samples from the flights listed in Table S1 had similar size-resolved features with varying magnitude in absorption (Figure S5).

These results are the first reported SEC-UV measurements of very fresh (0-5 h) field samples of biomass burning smoke, and they confirm some of the observations from field studies that measured more aged smoke, as well as laboratory studies that generated fresh or aged smoke. Previous studies that examined biomass burning BrC using SEC-UV have similarly concluded that fresh, less aged smoke contains a large fraction of lower MW absorbing species (Di Lorenzo et al., 2017; Wong et al., 2017; Lyu et al., 2021). In the examination of field samples, Di Lorenzo et al. (2017) collected ambient biomass burning aerosols that had been aged 10 to >72 h. They observed that low MW (<500 Da) chromophores contributed more to total absorption than higher MW (>500 Da) compounds in the least aged (10 to 15 h) biomass burning-derived filter extracts (Di Lorenzo et al., 2017). These findings resemble the absorption features of our FIREX-AQ samples, which span a plume age range from 0 to 5 hours. Wong et al. (2019) used SEC-UV to analyze filter extracts collected during fire seasons in Greece with atmospheric ages of 1 to ~70 h and observed that high MW species dominated total BrC absorption of the fresh and aged smoke. Differences between the FIREX-AQ aqueous samples and the results presented by Wong et al. (2019) can be driven by varying types of fuel emissions, photochemical conditions, meteorology, and differences in back trajectory analyses.

Two studies have applied SEC-UV analysis to lab-generated or lab-aged smoke samples. Wong et al. (2017) pyrolyzed dry hardwood and aged the samples from 0 to 10 h with UV light. They found that low MW chromophores dominated total absorption compared to high MW species, which is generally consistent with our observations. Lyu et al. (2021) generated biomass burning aerosol from laboratory combustion of boreal peat and also analyzed the aerosol by SEC-UV. Under the same SEC-UV separation conditions, the FIREX-AQ water aqueous samples parallel the findings of Lyu et al. (2021), with low MW BrC chromophores dominating total absorption for unaged fresh smoke and smoke aged between 0 to 5 h in the atmosphere.

**3.3 Comparison of SEC-UV and BrC-PILS absorption**

Online and offline absorption sampling are complimentary. The online sampling by the BrC-PILS provides continuous data with much higher time resolution (reported at 10 s), but it is limited to two measurements: water-soluble absorption as a function of wavelength and water-soluble organic carbon concentration. In contrast, offline samples can be examined using SEC-UV, $C_{18}$ chromatography, and other analytical techniques that are not feasible onboard an aircraft. During FIREX-AQ, the BrC-PILS measured online water-soluble absorption in the same aqueous flow that was collected for offline sampling. These are the only BrC samples whose absorption was measured online and then subsequently offline during FIREX-AQ. We observe differences between the online and offline samples. First, absorption by the offline SEC-UV at wavelengths greater than 300 nm did not exceed its detection limit. To facilitate comparison of absorption magnitudes, a power law fit was applied to the BrC-PILS absorption between 315–395 nm to extrapolate absorption to 300 nm (Figure 3). The total absorption by the SEC-UV measurements is approximately an order of magnitude greater than the BrC-PILS measurements at 300 nm (Figure 3). To determine if the differences could be attributed to differences in the diode array detector from the SEC-UV analysis and the BrC-PILS spectrometer, a standard solution of 4-nitrocatechol was run on both systems in pure water (without the SEC column in line and bypassing the PILS). At 350 nm, the agreement between the online, offline, and literature measurements of 4-nitrocatechol absorption was ±2.1% (Figure S13) (Hinrichs et al., 2016).

The comparison between the online and offline work presented here can be compared to previous non-co-located online-offline intercomparison studies. Di Lorenzo et al. (2017) compared offline absorption measurements of filter extracts by SEC-UV to PILS-LWCC online measurements during the Southern Oxidant and Aerosol Study (SOAS). Although the SEC-UV offline samples and online measurements during SOAS were not co-located, a median ratio (SEC-UV offline at 300 nm to PILS-LWCC online at 365 nm) of 0.9 and $r^2$ of 0.53 (Figure S12). Zeng et al. (2021) also present an online-offline absorption comparison of water-soluble BrC collected on board the NASA DC-8 aircraft during FIREX-AQ. Online absorption measurements by a LWCC and aqueous filter extracts injected onto a LWCC offline showed good agreement at 365 nm ($r^2 = 0.84$). The correlation suggested that the filter measurement of BrC is not significantly influenced by possible sampling artifacts associated with absorption of gases or evaporative loss of BrC components associated with filter collection (Zeng et al., 2021).

Differences observed in the FIREX-AQ aqueous samples indicate the necessity to
investigate a potential explanation for these inconsistencies. Since the online BrC-PILS and offline
SEC-UV samples represent the same samples, solubilization differences between aerosol
collection methods cannot explain the variability observed between our measurements. Although
reasonable agreement between online measurements and offline filter analyses has been
demonstrated (Di Lorenzo et al., 2017; Zeng et al., 2021), Resch et al. (2023) indicated that filter
extracts that are not refrigerated immediately or extracts that remain refrigerated for an extended
storage time are susceptible to compositional changes. For logistical reasons, our aqueous samples
were collected into polypropylene tubes in the field and were not immediately subjected to
controlled refrigeration or to the SEC-UV analysis. The greater absorption observed by the SEC-
UV analysis (Figure 3) could reflect the possible hydrolysis of oligomeric compounds stored in
aqueous solution resulting in an increase the intensity of precursor monomers as decomposition
products (Resch et al., 2023). We consider this, as well as other potential causes for the differences
in the next section.
**3.4 Investigating the impact of solvent effects, pH, and storage effects on absorption spectra**
The analysis of fresh biomass burning samples by SEC-UV may be affected by mobile
phase solvation effects, pH, and sample storage conditions. We analyze and discuss these variables
below and make recommendations for SEC-UV analysis. Plausible chemical structures of
chromophores responsible for BrC absorption have been identified and consist of conjugated
systems functionalized with hydroxyls, amines, nitro, carbonyls, and carboxylic acid groups
(Laskin et al., 2015; Hems et al., 2021; Lin et al., 2017; Fleming et al., 2020; Zeng et al., 2020b;
Hettiyadura et al., 2021; Marrero-Ortiz et al., 2019; De Haan et al., 2018; Ji et al., 2022). The
molecular complexities of BrC species may be susceptible to changes in the absorption spectra
depending on the analysis conditions.
First, we assess solvation effects due to changes in the mobile phase composition. The PILS
solubilizes BrC in pure water for the online measurements to facilitate absorption measurements
(Weber et al., 2001). In contrast, the mobile phase used for the offline SEC-UV analysis was a
mixture of acetonitrile and DIW with 25 mM ammonium acetate. Chromatographic packing
materials are often incompatible with pure water and require a mixture with an organic solvent to
elute compounds from the stationary phase or, in SEC separations, to prevent sorption to the
stationary phase. For this reason, chromatographic partitioning-based separations occur in
aqueous-organic mixtures, where the composition can be deliberately modified to optimize
interactions of the target molecules between the stationary phase and mobile phase. In SEC, non-
size exclusion interactions between the analyte and stationary phase are dominated by electrostatic
and hydrophobic interactions (Hong et al., 2012). If the analyte and stationary phase are identically
charged, ion exclusion effects can occur, resulting in an earlier elution time as the analyte is
prevented from entering the pores. If the analyte and stationary phase are oppositely charged,
adsorption can result, leading to a later elution time. Hydrophobic effects can occur if the analyte
interacts with hydrophobic sites of the column matrix (Hong et al., 2012). The purpose of adding
ammonium acetate to the mobile phase is to increase the ionic strength of the mobile phase and
facilitate ion-pairing, which suppresses electrostatic interactions between the stationary phase and
the polar and charged functional groups. The organic solvent used in our mobile phase was
acetonitrile, which has been shown to be unreactive towards typical BrC components and has been
recommended as an inert solvent for BrC extraction and analysis (Walser et al., 2008; Bateman et
al., 2008; Chen et al., 2022). Therefore, we do not expect the mobile phase to chemically alter BrC
compounds while effective at mitigating column stationary phase-analyte interactions.
While chemical changes caused by our mobile phase are unlikely, it is possible that other
solvent effects on absorption could be occurring. Effects of solvent on molecular absorption are
well established in the photochemistry literature (Lignell et al., 2014; Mo et al., 2017; Zheng et
al., 2018; Lyu et al., 2021; Chen et al., 2022; Dalton et al., 2023). The polarity of the solvent affects
the absorption wavelength by changing stabilization of the ground and/or excited states. With a
decrease in solvent polarity, (acetonitrile-water is less polar relative to pure water), this can lead
to a decrease in stabilization of the ground state of BrC compounds (such as 4-nitrocatchol), but
this effect is molecule-dependent. The impact of solvation on red and blue spectral shifting will
likely be several nanometers, which could contribute to the observed differences in the offline and
online absorption measurements. Previous work has shown that acetonitrile could disrupt π- π
interactions between BrC molecules, which could cause the liberation of adsorbed low MW BrC
chromophores from larger chromophores or disrupt BrC aggregates (Lyu et al., 2021). Smaller,
less conjugated systems typically absorb in the ultraviolet-blue wavelength region, and their π →
π* transition red shifts when more conjugated systems are fused together (Gorkowski et al., 2022).
Thus, we would expect absorption measurements in the presence of acetonitrile to be blue-shifted
relative to those in pure water. This represents a possible explanation for greater absorption
intensity at lower wavelengths measured in the offline SEC-UV analysis compared to the online
analysis.
Second, we assess the pH of the sample matrix, which is known to affect the absorption profile
of BrC compounds. Multiple studies have investigated the impact of pH on wavelength-dependent
absorption. For example, Phillips et al. (2017) directly adjusted the pH of SRFA and biomass-
burning derived aqueous extracts (with sodium hydroxide or hydrochloric acid) and observed no
measurable shift in the spectra to shorter or longer wavelengths; however they did observe that as
the pH increased, there was an increase in the magnitude of absorption, which was more
pronounced at higher wavelengths. The pH of the default mobile phase solution was 7.2, while the
pH of the deionized water solutions in the PILS was approximately 5 (due to carbon dioxide
dissolution). To investigate the impact pH has on BrC absorption, we measured several compounds
that have been shown to contribute to BrC absorption (4-nitrocatechol, vanillin, 7-
hydroxycoumarin, 4-hydroxy-3-methoxy cinnamaldehyde, and mixture of the four compounds)
under different solvent and pH conditions: DIW, DIW with 25 mM ammonium acetate, as well as
the mobile phase at pH 5, 7.2, and 9. When the matrix conditions have a pH greater than the $pK_a$
of the compound in question, the species will deprotonate, resulting in a shift to longer wavelengths
(Hinrichs et al., 2016). For compounds with a $pK_a$ between 5 and 9 (i.e., 4-nitrocatechol, 7-
hydroxycoumarin, vanillin), we observed this phenomenon (Figure S6). To assess the impact of
pH and mobile phase on a complex mixture, we also measured the absorption of a SRFA aqueous
solution and a FIREX-AQ aqueous sample with the abovementioned mobile phases (Figure 4 and
Figure S8). In contrast to the individual BrC compounds, no major changes in the spectral shape
were observed under different mobile phase conditions. To confirm these results, we measured the
absorption of SRFA in each solvent using a separate spectrophotometer (Figure S7). This suggests
that the $pK_a$ of the majority of functional groups in the absorbing compounds present were less
than 5 or above 9. Nitroaromatic compounds typically have $pK_a$ values between 5 and 8;
suggesting low levels of this class of compounds present in the aqueous samples. This observation
is comparable to the online BrC-PILS analysis; for aqueous absorption, Washenfelder et al. (2022)
observed the average absorption contribution at 365 nm of 4-nitrocatchol was less than 1.1 % and
the summed contribution to absorption by 2-nitrophenol, 4-nitrophenol, 4-nitrocatechol, 4-
nitroguaiacol, and 2,4-dinitrophenolate was less than 3.6 %. Since the absorption profile of SRFA
and the FIREX-AQ sample appear similar in all mobile phase conditions, we have no evidence
that pH of the mobile phase in the SEC separation conditions impacts the wavelength dependent
absorption of the FIREX-AQ aqueous samples.
Third, we assess the potential effect of storage on the aqueous samples measured by SEC-UV.
A recent study by Resch et al. (2023) observed that biomass burning-derived filter extracts stored
at temperatures above freezing may undergo compositional changes that can increase in signal for
various compounds. Hydrolysis reactions include converting alkenes to alcohols and esters to
carboxylic acids, and the breakdown of oligomers. The hydrolysis of oligomers such as dimer
esters stored in an aqueous solution can result in an increase in precursor monomers as
decomposition products leading to an increase in signal (Zhao et al., 2018; Resch et al., 2023).
Further, ammonium and alkylamines have been observed in high levels in biomass burning
aerosols (Di Lorenzo et al., 2018); aqueous reactions between dicarbonyls (e.g., glyoxal,
methylglyoxal) with ammonium and amines may also contribute to an increase in absorption
intensity at pH 4 to 7 (Powelson et al., 2014; Yang et al., 2023). The FIREX-AQ aqueous samples
had a pH of 5 and were stored at 4 °C for two years prior to analysis. Assuming they contained
dicarbonyl compounds and reduced nitrogenous species, it is possible reactions leading to products
that can contribute to greater absorption during storage occurred. To further investigate the impacts
of storage on a complex aqueous mixture, we measured the absorption spectra of two SRFA
solutions: one freshly made and one stored for one year at 4 °C. We observed an increase in
absorption in the aged SRFA solution, in which integrated absorption was 39 % higher than the
freshly-made solution. This same effect was also observed with SRHA solutions (Figure S14).
Thus, it is possible that processes during storage could have led to increased absorption measured
in the offline SEC samples.
Among the three processes discussed here, we conclude that the storage of aqueous extracts is
the most plausible explanation for the higher absorption observed in the offline samples from
FIREX-AQ. If hydrolysis reactions are occurring, we might expect this to impact the MW profile
(i.e., SEC elution times). We examined the MW profile of freshly-made and one year-aged SRFA
solutions (Figure 4C). The increase in absorption with storage does not measurably affect the
molecular size-resolved absorption of the mixtures. The same effect was observed for SRHA
(Figure S14). This demonstrates that any storage-induced changes in these complex mixtures of
organic molecules have a minimal impact on the molecular weight relative to the wide MW range
of the SEC column. The MW of the BrC species would have to change by ~ 100 Da to be noticeable
on the MW scale of our separation (250 Da to 75 kDa). Such a drastic change in MW is unlikely
the case in most hydrolysis reactions. Thus, our results above in which we broadly categorize MW
species to be less than or greater than 500 Da are likely robust. The SEC separation of the aqueous
samples signify that low MW (<500 Da) chromophores contribute more to total absorption than
higher MW (>500 Da), this finding is supported by previous SEC-UV analyses of BrC aged less
than 10 hrs (Di Lorenzo et al., 2017; Lyu et al., 2021). The consistent MW profiles between the
freshly-made and stored solutions of SRFA and SRHA reasonably suggest that storage did not
have a major impact on the MW of BrC.
**4. Conclusions and implications**
During FIREX-AQ, instruments onboard the NOAA Twin Otter aircraft sampled smoke
plumes from wildfires in the western United States with plume ages of 0 to 5 h. The BrC-PILS
measured water-soluble BrC absorption online and collected aerosol in aqueous solution for offline
SEC-UV analysis. The aqueous samples were collected during downwind plume transects and the
online data was collected continuously during inflight sampling. SEC-UV analysis shows that BrC
absorption was dominated by chromophores <500 Da. This finding is consistent with reports of
laboratory-generated fresh smoke samples. Integrated absorption at 300 nm from the SEC-UV
analysis was used to calculate trends in normalized BrC absorption as a function of plume age.
These trends were variable and did not show an exponential decay, which is consistent with
recently published results from the FIREX-AQ field campaign that examined normalized BrC
absorption trends for plumes over 0 to 10 h. Comparison of the online and offline analyses of the
same aqueous extracts reveals discrepancies, specifically higher absorption intensity and
absorption at lower wavelengths. These discrepancies between online and offline samples
demonstrate the importance of considering the conditions in which the absorption measurements
are made. The inconsistencies between the offline SEC-UV analysis and the online measurements
are not explained by pH or solvent effects, but may be due to reactions occurring during storage.
Although increases in absorption may occur during storage of aqueous solutions, it is less likely
to impact the MW of the FIREX-AQ BrC species. This highlights that BrC species are more stable
collected on filters rather than in aqueous solution and the importance of inter-comparison
absorption measurements by multiple methods.

**Acknowledgements**

We thank Carsten Warneke, Joshua Schwarz, James Crawford, and Jack Dibb for organizing the FIREX-AQ field campaign. We thank the NOAA Aircraft Operations Center for support during the field mission. L.A. was supported by a Mitacs Globalink Research Internship and an NSERC Discovery Grant. The FIREX-AQ project was supported by the NOAA Atmospheric Chemistry, Carbon, and Climate Program (AC4). We thank Robert Di Lorenzo and Trevor VandenBoer for helpful discussions. We thank three anonymous reviewers for their insightful feedback.

**Data Availability Statement**

The data used in the study are publicly available at https://www-air.larc.nasa.gov/missions/firex-aq/

**Competing Interests**

At least one of the (co-)authors is a member of the editorial board of Atmospheric Chemistry and Physics.

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

**Figure 1.** Ratio of absorption measured by SEC-UV at 300 nm to CO enhancement as a function of plume age for aqueous samples collected for six flights during FIREX-AQ 2019.

**Figure 2**. Absorption contribution at 300 nm of high (>500 Da), low (<500 Da), and unidentified molecular weight species for aqueous samples collected during the second flight leg on 21 Aug 2019.

**Figure 3.** Total absorption measured offline by the SEC-UV (at 300 nm) compared to the total absorption measured online by the BrC-PILS (extrapolated to 300 nm using a power-law fit). Each colour represents a different flight leg and each marker represents the integrated absorption at 300 nm for each aqueous sample measured by SEC-UV. The online BrC-PILS absorption measurement was averaged over the collection time of each aqueous sample. The error bars represent the total uncertainty in the online and offline measurements.

**Figure 4.** Absorption as a function of wavelength of (a) SRFA and (b) a FIREX-AQ aqueous sample collected on 28 Aug 2019 L3 with varying mobile phases. (c) Molecular weight profile of a freshly-made 15 µg/mL SRFA solution and the same solution one year later. The shaded region represents the coefficient of variation for absorption at each wavelength using n = 3 DIW.

**Figure 1.**

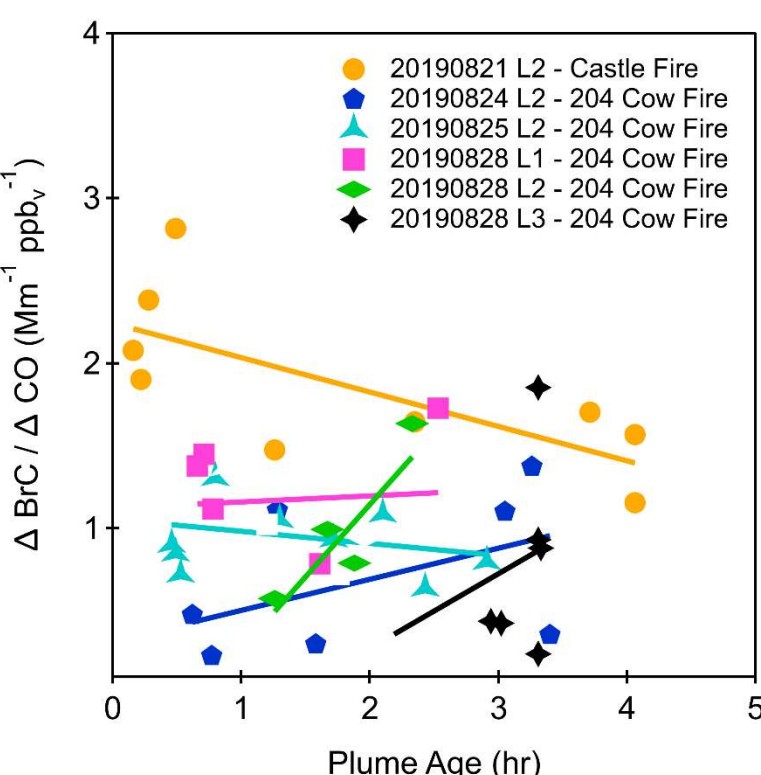

**Figure 2.**

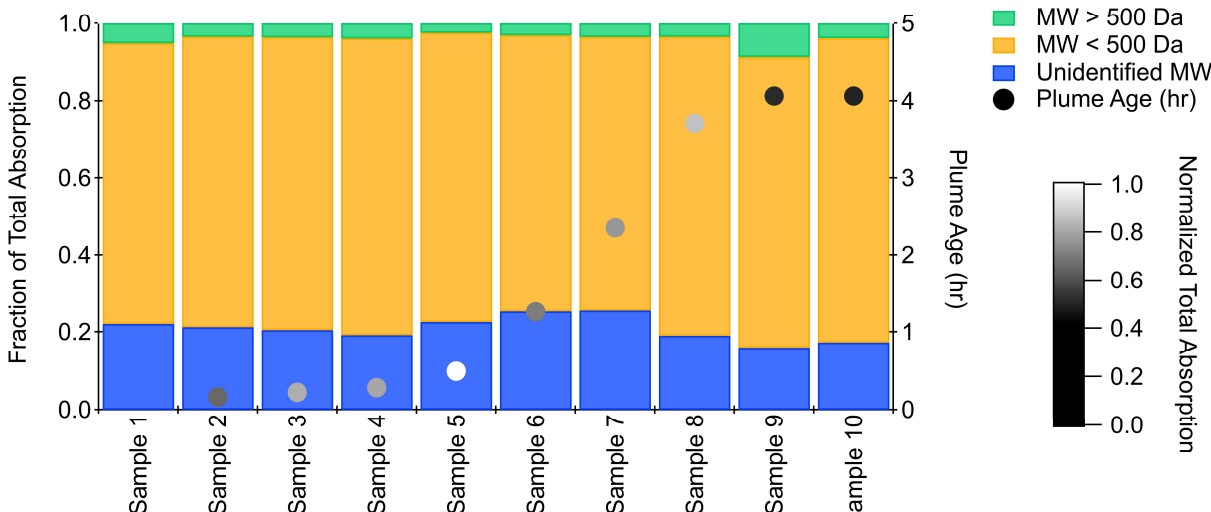


**Figure 3.**

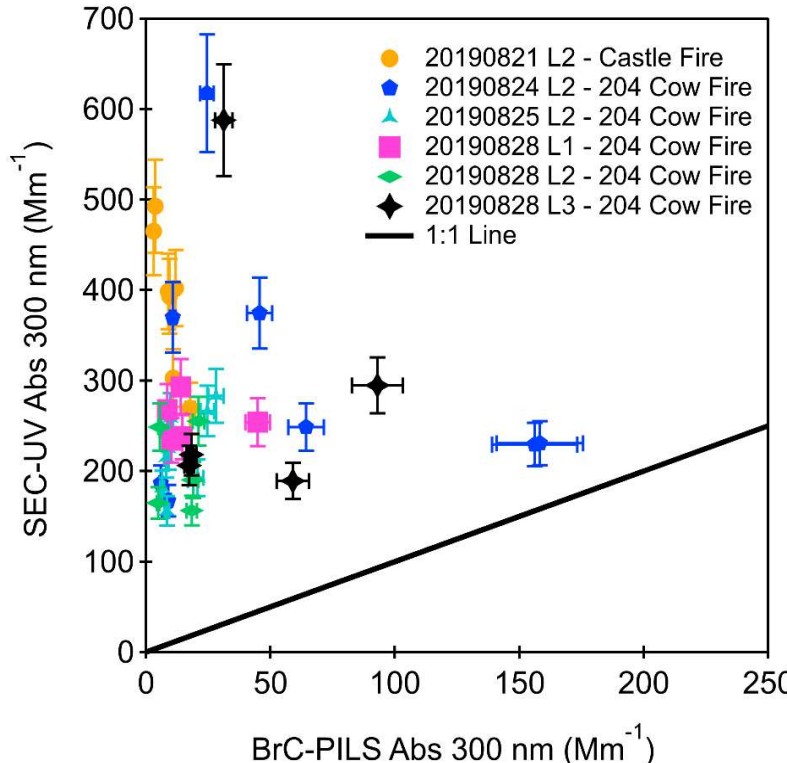


**Figure 4.**

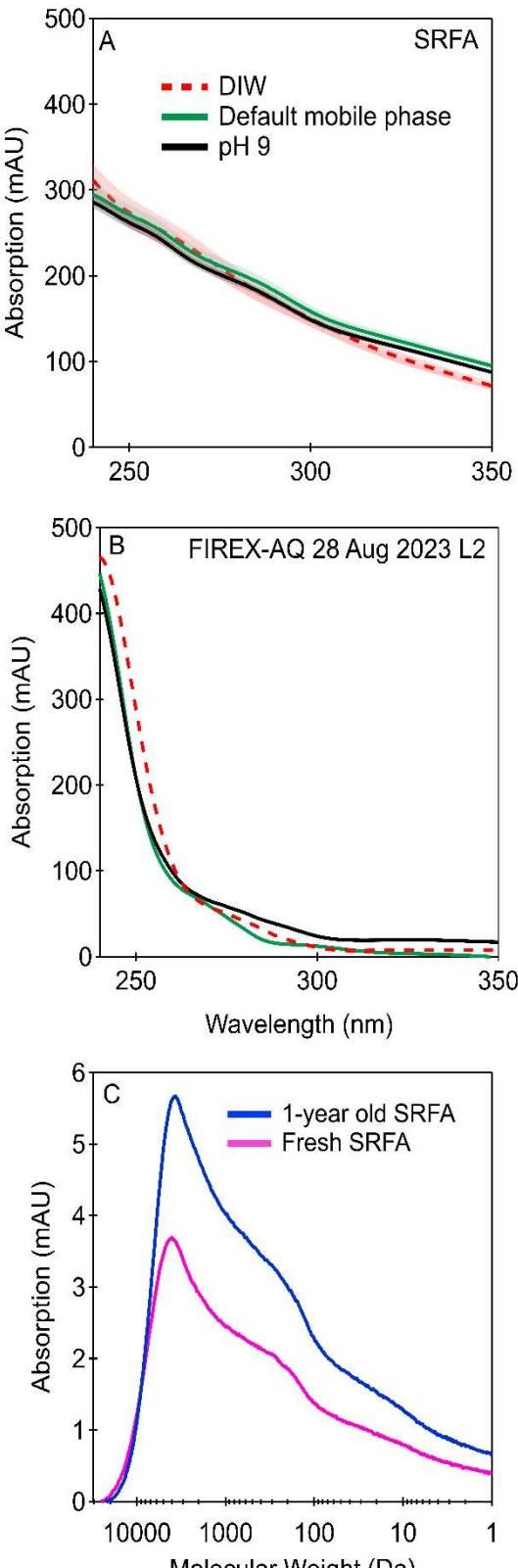