# Peer review of "Lisa Azzarello1, Rebecca A. Washenfelder2, Michael A. Robinson2,3, Alessandro Franchin2,3,4,"

_EGUsphere, 2023_

## Author Comment (AC3)

**Reviewer 3:**

**Review of "Characterization of water-soluble brown carbon chromophores from wildfire plumes in the western US using size exclusion chromatography" by Azzarello et al.**
**General comments: This manuscript presents results on a comparison of offline and online brown carbon (BrC) measurements for wildfire smoke collected from the Twin Otter aircraft during FIREX-AQ. The authors found that the BrC was dominated by smaller chromophores (<500 Da) and that there was not a consistent decrease in BrC with plume age (0-5 hours). They also found differences in the spectra between online and offline measurements and attribute this to solvent effects. Overall this is a very well written paper with a clear summary of the results. I have a few minor comments that should be addressed before this paper is accepted.**

We appreciate the detailed comments of the Reviewer. We have addressed each in detail below, where our responses are in highlighted blue and changes to the manuscript are indicated in **bold blue**. We have undertaken additional experiments and added new text and figures to the manuscript and SI. The changes are detailed below.

**Specific comments:**

1.  **It is noted that there isn't a consistent trend in BrC with plume age, as has been reported in other papers looking at these fires. Here the BrC is normalized to the average CO concentration. How much variation was observed in this CO mixing ratio during the sample collection period and how much variation was observed in general in the fire? How does this variation compare to previous studies using this normalization method?**

The CO mixing ratio that corresponds to the sampling duration of each aqueous sample was determined. The average CO mixing ratio and standard deviation for each flight is displayed in new Figure S9. To represent the variation observed in CO for each flight, the figure includes the coefficient of variation which represents the standard deviation as a percent of the mean; a greater coefficient of variation indicates more variability. Intercomparison of total measured absorption to CO was our focus since this is a standard way to assess the evolution of BrC while considering plume dilution.

[Figure]

**"Figure S9. Average CO measured and standard deviation for each flight and the blue markers denote the coefficient of variation, which represents the standard deviation as a percent of the mean."**

2. **Looking at Figure 1, the black markers have a line fit to them that appears to extend beyond the data shown in the figure. Is there data not shown in the figure?**

Figure 1 displays absorption at 300 nm measured by the SEC-UV for analysis normalized to CO as a function of plume age. The purpose of the linear fit is to represent whether BrC absorption was increasing or depleting as a function of plume age and to compare these findings to the literature. The linear fit for the flight denoted by the black markers does not extend past the data shown; the data shown in Figure 1 is not cut-off in any way.

3. **For the solvent effects, pH has also been demonstrated to play an important role in shifts in the absorption of BrC. What was the pH (estimated) of the water for the PILS analysis. Do you expect a difference compared to the pH that is found in the water collected for offline analysis? The idea of pH for organics in organic solvents is complicated, but the pH of the water in the online measurements may play a role in the differences observed.**

We agree that pH can play a role in absorption shifting. To investigate the impact the pH of the mobile phase has on wavelength dependent absorption, the ammonium acetate solution was adjusted to pH 5 and pH 9 prior to the addition of the acetonitrile. Several compounds that contribute to brown carbon absorption (4-nitrocatechol, vanillin, 7-hydroxycoumarin, 4-hydroxy-3-methoxy cinnamaldehyde, and mixture of the four compounds) and SRFA were analyzed in the following mobile phases: DIW, DIW with 25 mM ammonium acetate, the default mobile phase (not pH controlled), pH 5, and pH 9. There is red shifting when the pH was greater than the $pK_a$ of the compound. We acknowledge that numerous studies have investigated the impact directly acidifying and alkalinizing a SRFA sample have on absorption – there is a slight increase in absorption with increasing pH (Phillips et al., 2017). These results are shown in new Figure S6.

[Figure]

**"Figure S6. Absorption as a function of wavelength measured in various mobile phases of (a) 4-nitrocatechol (b) 4-hydroxy-3-methoxy cinnamaldehyde (c) vanillin (d) 7-hydroxycoumarin and (e) a mixture of the four compounds by a diode array detector. The default mobile phase consists of 1:1 mixture of acetonitrile and DIW with 25 mM ammonium acetate. The purple and black traces represent pH controlled mobile phases, where the pH of the ammonium acetate solution was adjusted to pH 5 and pH 9 prior to the addition of the acetonitrile."**

The pH of the PILS water was approximately 5. Since the aqueous samples flowed through the liquid waveguide capillary cell prior to its collection in the polypropylene tubes, we do not expect a difference in pH of between the online analysis and aqueous samples. To assess the impact of pH and mobile phase on a complex mixture, we also measured the absorption of a SRFA aqueous solution and a FIREX-AQ aqueous sample with the abovementioned mobile phases (Figure 4 and Figure S8). In contrast to the individual BrC compounds, no major changes in the spectral shape were observed under different mobile phase conditions. To confirm these results, we measured the absorption of SRFA in each solvent using an Agilent spectrophotometer (Figure S7).

[Figure]

**"Figure 4. Absorption as a function of wavelength of (a) SRFA and (b) a FIREX-AQ aqueous sample collected on 28 Aug 2019 L3. The default mobile phase consists of 1:1 mixture of acetonitrile and DIW with 25 mM ammonium acetate. The purple and orange traces represent**

**pH controlled mobile phases, where the pH of the ammonium acetate solution was adjusted to pH 5 and pH 9 prior to the addition of the acetonitrile."**

[Figure]

**"Figure S7. Absorption as a function of wavelength of SRFA measured using a Agilent 8453 UV-visible Spectroscopy System. A solution of 15 µg/mL SRFA was diluted by 50 % by the mobile phase and then transferred to the cuvette prior to measurement. The default mobile phase consists of 1:1 mixture of acetonitrile and DIW with 25 mM ammonium acetate. The purple and orange traces represent pH controlled mobile phases, where the pH of the ammonium acetate solution was adjusted to pH 5 and pH 9 prior to the addition of the acetonitrile."**

[Figure]

**"Figure S8. Absorption as a function of wavelength of SRFA measured in a 25 mM ammonium acetate solution and in a mobile phases controlled to pH 5."**

**4. Solvent effects can play a role in the position of the absorption, but I've seen less discussion around changing the overall intensity of the absorption. Do you think that the increase for the offline measurements can be attributed to this, or are there other possible reasons for that increase?**

The offline SEC-UV analysis and the online absorption measurements by the BrC-PILS were converted to Mm$^{-1}$, a common unit used to express the magnitude of BrC absorption. The total absorption by the SEC-UV measurements is approximately an order of magnitude greater than the BrC-PILS measurements at 300 nm. We are confident that the optical system and the diode array detector can measure absorption that are within agreeance. For example, we compared absorption measurements of 4-nitrocatechol by the LWCC of the BrC-PILS and the diode array detector converted to absorption cross section, showing good good agreement (Figure S13 below).

[Figure]

**"Figure S13. Absorption cross section of 4-nitrocatechol measured by injection onto the LWCC of the BrC-PILS and onto the diode array detector of the SEC-UV set-up compared to Hinrichs et al. 2016."**

The investigation of pH impact on absorption prompted the investigation of other plausible explanations to describe the discrepancies between the online BrC absorption measurements and offline SEC-UV measurements. Major differences between the online and offline analyses are the use of acetonitrile for the chromatographic separation and that the aqueous samples were stored prior to the SEC-UV analysis.

Therefore, in considering possible pH, solvent, and storage effects, section 3.4 was retitled and rewritten:

Section 3.4 was titled: 3.4 Solvents affect the measured absorption spectra

This was changed to:

[revised manuscript text omitted]

---

## Author Response (AR1)

**Reviewer 1:**

The manuscript examines water-soluble light-absorbing compounds in biomass burning fire plumes using on-line particle into liquid sampler (PILS) and offline size exclusion chromatography (SEC), both coupled to spectrophotometric detectors. The main messages of the manuscript are: (1) absorption coefficient of the BBOA particles sometimes increases and sometimes decreases with the plume age; (2) PILS and SEC data do not agree with each other due to effects of solvents on the absorption spectra of the analyzed chromophores. While the manuscript is potentially publishable, I have two major comments about the manuscript that will likely require a major revision or an even more drastic action.

We appreciate the detailed comments of the Reviewer. We have addressed each in detail below, where our responses are in highlighted blue and changes to the manuscript are indicated in **bold blue**. In summary, we agree with the comments of the Reviewer and have performed several new experiments to explore the effects of solvent composition on brown carbon absorption. This is reflected in new text and figures in both the manuscript and SI, described below.

1). The section describing the solvent effects will need to be significantly revised. The authors have misinterpreted the change in the spectrum of nitrocatechol as the effect of solvent polarity. Instead, this is quite simply an acid-base equilibrium between nitrocatechol (C6H5NO4, absorption peak at around 350 nm) and its anion (C6H4NO4-, absorption peak at around 450 nm). Common nitrophenols have pKa of the order of 7, leading to large differences in the absorption spectra recorded during LC separation using an acidified and non-acidified eluent. For example, see Figure 5 of Cornard et al. (2005), https://doi.org/10.1016/j.chemphys.2004.09.020, for the comparison of absorption spectra of the nitrocatechol and its anion. Also, see Figure S2 in Lin et al. (2017), https://doi.org/10.1021/acs.est.7b02276, which shows how spectra of eluted nitrocatechol and other nitrophenols change depending on the eluent pH.

The reviewer is correct regarding the pH effect on nitrocatechol. Nitrocatechol, having a pKa value of 6.84, is deprotonated in these conditions, resulting in a red shift. We have removed the text in section 3.4 that incorrectly explains the reason why we observed a red shift for 4-nitrocatetchol. We have moved and edited Figure 4b and moved it to the SI as part of Figure S6, where we show the wavelength dependent absorption of several compounds known to contribute to BrC absorption under various mobile phase conditions. The details of Figure 4, S6, and rewriting of section 3.4 are explained below.

From what I can gather from Figure 4 in this manuscript, the ammonium acetate buffer that the authors used for their SEC ACN+buffer experiments was sufficiently basic to significantly deprotonate nitrocatechol. In contrast, in their DIW only experiment, nitrocatechol was only partly deprotonated (there is a shoulder there corresponding to the 450 nm band of the anion but the majority of the nitrocatechol is not deprotonated). Was the buffer prepared to provide buffering at pH 5 or pH 9 in this work? I presume it is the latter. For more on this buffer and its dual pH buffering nature, I would recommend reading Konermann (2017), https://doi.org/10.1007/s13361-017-1739-3.

Ammonium acetate was dissolved in water and then combined with acetonitrile, which has a pH of 7.2. Ammonium acetate dissolved in water has an inherent buffering capacity between pH 3.75 to 5.75 and pH 8.25 to 10.25 (Konermann, 2017); however, with the addition of acetonitrile, the buffering ranges shifts to approximately 5.5±1 and pH 9 ±1 (Subirats et al., 2009). The pKa values of previously identified BrC compounds are within the inherent buffering range so we did not pH control the mobile phase.

The ammonium acetate solution was not pH adjusted and was inaccurately called a buffer in the manuscript. We should have referred to it as a mobile phase modifier. We have corrected this statement in sections 2.3, 3.4, and throughout the manuscript and SI:

Line 158 to 161: The isocratic method was run using a 1:1 mixture of acetonitrile and a buffer solution consisting of 18.2 MΩ·cm deionized water with 25 mM ammonium acetate at a flow rate of 1 mL min$^{-1}$ and a sample injection volume of 100 µL.

This sentence was changed to:

**Line 158 to 161: "The isocratic method was run using a 1:1 mixture of acetonitrile and 25 mM ammonium acetate at a flow rate of 1 mL min$^{-1}$ and a sample injection volume of 100 µL."**

In the SI:

Figure S4: Single-wavelength chromatogram at 250 nm of an aqueous sample run with equal parts buffer solution and methanol (black) and equal parts buffer solution and acetonitrile (orange).

This caption was changed to:

**"Figure S4: Single-wavelength chromatogram at 250 nm of an aqueous sample run with equal parts 25 mM ammonium acetate solution and methanol (black) and equal parts 25 mM ammonium acetate solution and acetonitrile (orange)."**

The following text was added to the SI:

**"The default mobile phase used was equal parts acetonitrile and DIW with 25 mM ammonium acetate. When ammonium acetate is dissolved in water, sub-stoichiometric acidification converts acetate to acetic acid producing conditions that can stabilize pH at 4.75 (Konermann, 2017). The alkalinization of ammonium acetate solution generates $NH_3$ via the depronotation of $NH_4^+$, creating buffering capacity around the $pK_a$ of ammonium (9.25). Therefore, ammonium acetate dissolved in water has an inherent buffering capacity in acidic (pH at 4.75±1) and basic ranges (9.25±1) (Konermann, 2017). The addition of acetonitrile to ammonium acetate dissolved in water reduces the buffer capacity and shifts the buffering ranges of ammonium acetate dissolved in water to approximately pH 5.5±1 and pH 9 ±1 (Subirats et al., 2009). The purpose of the addition of the ammonium acetate to the mobile phase was to minimize electrostatic interactions between the compounds and the stationary phase of the column. This has proven effective in previous SEC-UV analyses of biomass burning derived samples investigating MW properties of fresh and aged BrC (Di Lorenzo et al., 2017; Wong et al., 2017; Lyu et al., 2021). If the electrostatic interactions are negligible, SEC separation is based on hydrodynamic volume, which is a function of MW and the density of the compounds (Pelekani et al., 1999). In Figure S6, there is a red shift when mobile phase conditions have a pH greater than the $pK_a$ of the single compound. However, Figure S7 shows that the wavelength-dependent absorption of SRFA looks**

**similar under all mobile phase conditions. This indicates that we do not anticipate pH impacting wavelength-dependent absorption in the SEC-UV analysis.”**

**2). The strong pH dependence of the absorption spectra of nitrophenols (and some other brown carbon compounds) and the different acidities of working solvents used for the PILS and SEC portions of this work, make it very hard to faithfully compare the results obtained by these two methods. I presume that the complete lack of correlation between the two methods in Figure 3 must be at least in part due to these solvent acidity effects. Broadly speaking, this manuscript shows that choosing an inappropriate solvent for the measurements will lead to questionable results. Is this self-evident conclusion really worth publishing? Would the agreement be better if a more acidic buffer was used for the SEC portion of the work? Given that the atmospheric particles that are commonly acidic, why was a basic buffer selected for the separation? In my opinion these questions need to be carefully addressed before the manuscript can proceed to a publication. Additional experiments (and possibly a full re-analysis of samples with a different solvent for SEC) may be necessary to address these questions.**

We agree with the Reviewer that an investigation on the impact of pH was warranted, and we have performed additional experiments to examine this question. To investigate the impact of mobile phase pH on wavelength dependent absorption, we adjusted the ammonium acetate solution to pH 5 and pH 9 prior to the addition of the acetonitrile. We selected four compounds that contribute to brown carbon absorption (4-nitrocatechol, vanillin, 7-hydroxycoumarin, 4-hydroxy-3-methoxy cinnamaldehyde, and mixture of the four compounds) and SRFA to run in the following mobile phases: DIW, DIW with 25 mM ammonium acetate, the default mobile phase (pH of 7.2), pH 5, and pH 9. As expected, we observed red shifting when the pH was greater than the $pK_a$ of the compound. We acknowledge that numerous studies have investigated the impact directly acidifying and alkalinizing a SRFA sample have on absorption – there is a slight increase in absorption with increasing pH (Phillips et al., 2017).

To assess the impact of pH and mobile phase on a complex mixture, we also measured the absorption of a SRFA aqueous solution and a FIREX-AQ aqueous sample with the abovementioned mobile phases (Figure 4 and Figure S8). In contrast to the individual BrC compounds, no major changes in the spectral shape were observed under different mobile phase conditions. To confirm these results, we measured the absorption of SRFA in each solvent using an Agilent spectrophotometer (Figure S7).

The investigation of pH impact on absorption prompted the investigation of other plausible explanations to describe the discrepancies between the online BrC absorption measurements and offline SEC-UV measurements. Major differences between the online and offline analyses are the use of acetonitrile for the chromatographic separation and that the aqueous samples were stored prior to the SEC-UV analysis. These results are now described in the revised manuscript. Section 3.4 was retitled and re-written, and a description of the new methods was added as section 2.4.

**“2.4 Absorption in different mobile phases**

**To assess the impact of pH and mobile-phase composition on wavelength-dependent absorption, the ammonium acetate solution was adjusted to pH 5 and pH 9 with acetic acid and ammonium hydroxide, respectively, prior to combining with acetonitrile. A 15 μg/mL in DIW solution of Suwannee River Fulvic Acid (SRFA II; International Humic Substances Society, Saint Paul, MN, USA) and a FIREX-AQ aqueous sample were injected onto the diode array detector without the SEC column in line with the following mobile phases: DIW only; 25 mM ammonium acetate solution; the default mobile phase (described in Sect. 2.3); 25 mM ammonium acetate**

solution adjusted to pH 5; and 25 mM ammonium acetate solution adjusted to pH 9. Solutions of 4-nitrocatechol, 4-hydroxy-3-methoxy cinnamaldehyde, vanillin, and 7-hydroxycoumarin in DIW with concentrations of $3.9×10^{-8}$, $3.4×10^{-8}$, $3.9×10^{-8}$, $3.7×10^{-8}$ mol/mL, respectively, were prepared and injected onto the diode array detector to observe differences in their absorption profiles. To confirm the diode array detector results, measurements of the SRFA solution were also made with UV-visible spectroscopy (8453; Agilent Technologies, Santa Clara, CA, USA) where the solution was mixed (1:1 ratio) with the various mobile phases prior to transferring to a cuvette for absorption measurements (Figure S7)."

Section 3.4 was titled:
Line 287: 3.4 Solvents affect the measured absorption spectra

This was changed to:

[revised manuscript text omitted]

**nitrocatechol (b) 4-hydroxy-3-methoxy cinnamaldehyde (c) vanillin (d) 7-hydroxycoumarin and (e) a mixture of the four compounds by a diode array detector. The default mobile phase consists of 1:1 mixture of acetonitrile and DIW with 25 mM ammonium acetate. The purple and black traces represent pH controlled mobile phases, where the pH of the ammonium acetate solution was adjusted to pH 5 and pH 9 prior to the addition of the acetonitrile."**

[Figure]

**"Figure 4. Absorption as a function of wavelength of (a) SRFA and (b) a FIREX-AQ aqueous sample collected on 28 Aug 2019 L3 with varying mobile phases. (c) Molecular weight profile of a freshly-made 15 µg/mL SRFA solution and the same solution one year later. The shaded region represents the coefficient of variation for absorption at each wavelength using n = 3 DIW."**

[Figure]

**"Figure S7. Absorption as a function of wavelength of SRFA measured using a Agilent 8453 UV-visible Spectroscopy System. A solution of 15 μg/mL SRFA was diluted by 50 % by the mobile phase and then transferred to the cuvette prior to measurement. The default mobile phase consists of 1:1 mixture of acetonitrile and DIW with 25 mM ammonium acetate. The purple and orange traces represent pH controlled mobile phases, where the pH of the ammonium acetate solution was adjusted to pH 5 and pH 9 prior to the addition of the acetonitrile."**

[Figure]

**"Figure S8. Absorption as a function of wavelength of SRFA measured in a 25 mM ammonium acetate solution and in a mobile phases controlled to pH 5."**

[Figure]

**"Figure S14. Size separation of a fresh SRHA solution which was then re-run 20 months later."**

**References**

[revised manuscript text omitted]

Reviewer 2:

**This paper focuses on brown carbon determined in samples collected from a PILS during aircraft measurements as part of the Twin Otter component of the FIREX study. The BrC reported here is the light absorption at 300 nm, a wavelength somewhat lower than 365 nm, which is what is typically reported for BrC. This is stated to be due to extensive dilution of the sample. These offline samples are run through SEC to assess the molecular weight of the chromophores, but the focus seems to be mainly on the change in the spectral properties of BrC with different solvents needed for the SEC analysis. By comparing BrC from an online instrument (PILS) to SEC, the latter involving the addition of acetonitrile and ammonium acetate buffer, differences are observed. They conclude that solvents can affect the spectral properties of BrC. Other studies have noted that solution pH and organic solvents can cause this issue. A major limitation in this work is that the authors never compared the un-altered offline samples to the online samples to make sure that dilution or differences in sample handler did not cause spectral shift issues. There are discrepancies between what this paper reports and other research comparing solvent and online measurements of BrC that could be discussed in more detail. Finally, the authors might consider what their reference BrC measurement really is; what is considered the correct BrC measurement that reflects the characteristics of actual particles?**

We appreciate the detailed comments of the Reviewer. We have addressed each in detail below, where our responses are in highlighted blue and changes to the manuscript are indicated in **bold blue**. In response to comments, we have conducted additional experiments and expanded/modified both the manuscript and the SI. Our changes are detailed below.

**Specific Comments.**

**Typo line 102; edit: attribute assign.**

"attribute" has been deleted from the sentence.; the sentence has been corrected to:

> **Line 102: "We compare the total absorption measured in online and offline samples and assign the BrC absorption to different MW classes."**

**The BrC of this study is defined somewhat differently than most other studies. Line 175 – to 177. This method focuses on light absorption in the 250 to 300 nm range since it is stated that the samples were too dilute to detect absorption above blanks at higher wavelengths. Does this affect the analysis? Is this a spectral range where BrC is optically important, from a climate/radiative forcing perspective, if not why concerned about it?**

In the literature, common wavelengths reported for BrC absorption are greater than 300 nm. Because of the low concentration in the samples, the SEC-UV analysis did not observe absorption above 300 nm and therefore focused on absorption at 250 nm. The purpose of the SEC-UV analysis is not specifically to quantify the absorbance, but rather to provide molecular information that can aid in our understanding of the composition of BrC. The absorption measurements obtained by the SEC-UV analysis are useful to interpret molecular characteristics through size separation.

**Analysis of BrC exclusively based on these low wavelengths is somewhat unusual. The authors have access to the online water-soluble species absorption data from the PILS. Have they compared the methods, ie run the offline samples prior to any alteration (addition of the buffer and acetonitrile) for the SEC analysis and compared the data to the PILS? This would provide a baseline, addressing possible issues such as differences in dilution, sampling handling, etc.**

To measure optical properties in the absence of stationary phase, organic solvent, and mobile phase additives, a FIREX-AQ aqueous sample, Suwanee River fulvic acid (SRFA), and a mixture of several compounds was injected onto the diode array detector in pure DIW mobile phase without the SEC column inline (red trace in Figure 4a below). In the absence of size-exclusion separation, the wavelength dependent absorption showed little difference between pure DIW and mobile phases that contain organic solvent and additives. This suggests that the composition of the SRFA and the aqueous sample was unaltered by the mobile phase. However, an organic solvent and additives are required in SEC separation to mitigate column-analyte interactions.

To address why an organic solvent and ammonium acetate are required in a SEC separation, we added the following text to the SI:

**"The default mobile phase used was equal parts acetonitrile and DIW with 25 mM ammonium acetate. When ammonium acetate is dissolved in water, sub-stoichiometric acidification converts acetate to acetic acid producing conditions that can stabilize pH at 4.75 (Konermann, 2017). The alkalinization of ammonium acetate solution generates $NH_3$ via the depronotation of $NH_4^+$, creating buffering capacity around the pka of ammonium (9.25). Therefore, ammonium acetate dissolved in water has an inherent buffering capacity in acidic (pH at 4.75$\pm$1) and basic ranges (9.25$\pm$1) (Konermann, 2017). The addition of acetonitrile to ammonium acetate dissolved in water reduces the buffer capacity and shifts the buffering ranges of ammonium acetate dissolved in water to approximately pH 5.5$\pm$1 and pH 9 $\pm$1 (Subirats et al., 2009). The purpose of the addition of the ammonium acetate to the mobile phase was to minimize electrostatic interactions between the compounds and the stationary phase of the column. This has proven effective in previous SEC-UV analyses of biomass burning derived samples investigating MW properties of fresh and aged BrC (Di Lorenzo et al., 2017; Wong et al., 2017; Lyu et al., 2021). If the electrostatic interactions are negligible, SEC separation is based on hydrodynamic volume, which is a function of MW and the density of the compounds (Pelekani et al., 1999). In Figure S6, there is a red shift when mobile phase conditions have a pH greater than the pKa of the single compound. However, Figure S7 shows that wavelength dependent absorption of the SRFA looks similar under all mobile phase conditions. This indicates that we do not anticipate pH impacting wavelength dependent absorption in the SEC-UV analysis."**

**It seems that no offline measurement of BrC was made in this study without the addition of other solvents to the water samples? Is that correct? This should be clarified in the methods section. A plot like Fig 3 involving a direct comparison between the PILS and collected vials would be very informative and help interpret Fig 3.**

The Reviewer is correct. Chromatographic separations are not compatible with pure aqueous solutions, so any offline separation requires modification of the solvent. The FIREX-AQ aqueous samples collected into polypropylene tubes by the BrC-PILS were analyzed using the SEC-UV method where the mobile phase consisted of 1:1 mixture of a 25 mM ammonium acetate solution added to acetonitrile.

We have clarified this by modifying the text in section 2.3:

Text changed from:

Line 158 to 161: The isocratic method was run using a 1:1 mixture of acetonitrile and a buffer solution consisting of 18.2 MΩ·cm deionized water with 25 mM ammonium acetate at a flow rate of 1 mL min$^{-1}$ and a sample injection volume of 100 μL.

To:

**Line 158 to 161: "The isocratic method was run using a 1:1 mixture of acetonitrile and 25 mM ammonium acetate at a flow rate of 1 mL min$^{-1}$ and a sample injection volume of 100 μL."**

Lines 164 to 165 clarify that no post-sampling processing were required:

**Line 164 to 165: The aqueous samples collected by the BrC-PILS did not require post-sampling processing and were injected onto the SEC column under mobile phase flow to the diode array detector.**

We analyzed SRFA and one FIREX-AQ sample directly without a chromatographic separation. For these samples, we were able to examine pure aqueous solvent. We also investigated the impact ammonium acetate, acetonitrile, and pH has on the wavelength dependent absorption of this BrC proxy.

These experiments are described in the new section 2.4.

**"2.4 Absorption in different mobile phases**

**To assess the impact of pH and mobile-phase composition on wavelength-dependent absorption, the ammonium acetate solution was adjusted to pH 5 and pH 9 with acetic acid and ammonium hydroxide, respectively, prior to combining with acetonitrile. A 15 μg/mL in DIW solution of Suwannee River Fulvic Acid (SRFA II; International Humic Substances Society, Saint Paul, MN, USA) and a FIREX-AQ aqueous sample were injected onto the diode array detector without the SEC column in line with the following mobile phases: DIW only; 25 mM ammonium acetate solution; the default mobile phase (described in Sect. 2.3); DIW adjusted to pH 5; and DIW adjusted to pH 9."**

The results are described in revised text of section 3.4.

**"To assess the impact of pH and mobile phase on a complex mixture, we also measured the absorption of a SRFA aqueous solution and a FIREX-AQ aqueous sample with the abovementioned mobile phases (Figure 4 and Figure S8). In contrast to the individual BrC compounds, no major changes in the spectral shape were observed under different mobile phase conditions. To confirm these results, we measured the absorption of SRFA in each solvent using a separate spectrophotometer (Figure S7)."**

[Figure]

**"Figure 4. Absorption as a function of wavelength of (a) SRFA and (b) a FIREX-AQ aqueous sample collected on 28 Aug 2019 L3 with varying mobile phases. (c) Molecular weight profile of a freshly-made 15 µg/mL SRFA solution and the same solution one year later. The shaded region represents the coefficient of variation for absorption at each wavelength using n = 3 DIW."**

[Figure]

**"Figure S7. Absorption as a function of wavelength of SRFA measured using a Agilent 8453 UV-visible Spectroscopy System. A solution of 15 μg/mL SRFA was diluted by 50 % by the mobile phase and then transferred to the cuvette prior to measurement. The default mobile phase consists of 1:1 mixture of acetonitrile and DIW with 25 mM ammonium acetate. The purple and orange traces represent pH controlled mobile phases, where the pH of the ammonium acetate solution was adjusted to pH 5 and pH 9 prior to the addition of the acetonitrile."**

[Figure]

**"Figure S8. Absorption as a function of wavelength of SRFA measured in a 25 mM ammonium acetate solution and in a mobile phases controlled to pH 5."**

Lines 180 to 203, one could also include the findings from the WeCan study, ie: Sullivan, A., R. P. Pokrhet, Y. Shen, S. M. Murphy, D. W. Toohey, T. Campos, J. Lindaas, E. V. Fischer, and J. L. Collett (2022), Examination of Brown Carbon Absorption from Wildfires in the Western U.S. During the WE-CAN Study, *Atmos Chem Phys*, *22*, 13389-13406.

The findings described in this paper have been added:

**Lines 226 to 230: "In another study of fresh plumes, aircraft based measurements during the Western Wildfire Experiment for Cloud Chemistry, Aerosol Absorption and Nitrogen (WE-CAN; Sullivan et al., 2022) investigated the evolution of water-soluble BrC at 405 nm normalized to CO and observed BrC depletion with a smoke age of < 2 h, and PILS water-soluble BrC absorption that broadly remained stable for a smoke age up to 9 h (Sullivan et al., 2022)."**

In Fig 2, define what absorption means. I assume it is the same as in Fig 1, light absorbance from the LWCC at 300 nm.

The caption for Figure 2 was changed from:

Figure 2. Absorption contribution of high (>500 Da), low (<500 Da), and unidentified molecular weight species of aqueous collected during the second flight leg on 21 Aug 2019.

To:

**"Figure 2. Absorption contribution at 300 nm of high (>500 Da), low (<500 Da), and unidentified molecular weight species of aqueous samples collected during the second flight leg on 21 Aug 2019."**

Related to the above discussion on lack of comparison between un-altered offline samples and PILS. Lines 282-283. This implies that there are no published comparisons between online and offline water-soluble BrC measurements. Is this true? I suggest a literature search. See for example Fig 8 in Zeng et al. ()

We agree with the Reviewer; there was online sampling that corresponded to filter sampling for offline absorption measurements. However, the work presented here is the first analysis to measure online water-soluble BrC absorption properties with a corresponding offline SEC-UV analysis.

The comparison presented by Zeng et al. 2021 have been added to section 3.3.

**Lines 306 to 312: " Zeng et al. 2021 also present an online-offline absorption comparison of water-soluble BrC collected on board the NASA DC-8 aircraft during FIREX-AQ. Online absorption measurements by a LWCC and aqueous filter extracts injected onto a LWCC offline showed good agreement at 365 nm ($r^2$ = 0.84). The correlation suggested that the filter measurement of BrC is not significantly influenced by possible sampling artifacts associated with absorption of gases or evaporative loss of BrC components associated with filter collection (Zeng et al., 2021)."**

Related to the above is the question of published results comparing online vs solvent-extracted offline analysis, and the assertion that methanol can lead to artifacts (lines 307-309). It is noteworthy that

this has not been seen in biomass burning plumes measured during FIREX, see Fig 5a (and supplemental Fig S5) in Zeng et al, https://doi.org/10.5194/acp-22-8009-2022. Maybe other solvents, such as those used in the SEC analysis, produce substantial changes; how does one explain these discrepancies?

We agree with the Reviewer—it is interesting that methanol-induced effects were not observed by Zeng et al. (2022). This is in contrast to results from other studies and suggests that reactive solvent effects may be mixture dependent. However, there is no evidence from any studies that acetonitrile can induce changes in BrC. Chen et al., 2022 investigated the impact methanol and acetonitrile have on BrC absorption and demonstrated that methanol could react with chromophores that have conjugated carbonyl functionalities (for example: phthalic anhydride, maleic anhydride, and maleimide) and these reactions alter the absorption properties of BrC. (Kristensen and Glasius, 2011) also observed that carboxylic acids (such as pinonic acid and adipic acid) could generate methyl esters after methanol extraction but remained intact when acetonitrile was used. This is supported by our analysis of SRFA in various mobile phases (including a matrix matched mobile phase), and we did not observe changes in the absorption spectra of SRFA This is displayed in the revised Figure 4 (above).

A final point to consider is the idea that there is an ideal sampling method to measure BrC that does not alter the aerosol particle from its native state and so measured actual characteristics of BrC in an ambient particle.
Taking pH as an example, the issue raised by the other reviewer, maybe solvent extractions give the best option since pH can be adjusted to that expected for the particles, whereas online methods, such as the PAS, dry the particles to reduce artifacts, which can drastically change particle pH. The question is, can the true spectral properties of an ambient particle be measured without alteration? If not, what is the reference that things should be compared to, or should the focus be on noting and understanding factors that can affect spectral properties? In this case, it seems the water-soluble BrC is the reference, but at what pH (dilute solution in equilibrium with air pH~5), which raises the issue if the PILS and vials collected gave similar results, as noted above.

The Reviewer raises an excellent point. Identifying a reference that should be compared to when analyzing offline and online absorption measurements would be ideal; however, it is quite complicated. Many differences exist between established offline methodologies, including filter type, extraction solvent, and sequential extraction in DIW, then in an organic solvent, and varying spectrometer sensitivities. Investigating how these variables potentially alter BrC species at a molecular or aggregate level is important. Controlling the pH of the offline measurement conditions is an important idea; it encourages consideration of the $pK_a$ values of BrC compounds. Although aerosols are typically acidic, acidity is a range, and estimating what the pH should be may alter BrC species in ways that are not representative of the aerosol pH. Since BrC compounds have a range of $pK_a$ values, controlling the pH may not necessarily accurately describe the absorption properties of BrC. Offline analysis subjects samples to post-collection processing, and it is important that each step be investigated for possible implications.

The offline analysis presented here did not require post-collection processing since we collected the outflow from the LWCC; therefore, we decided to investigate how various mobile phases could influence the wavelength-dependent absorption. The results are displayed in Figures 4, S6, and S7. Overall, we determined that wavelength-dependent absorption was minimally impacted, which provided insight into the molecular complexities of BrC species. For instance, it may be possible that depending on the configuration of the BrC aggregates, the solvent may not have access to heteroatoms or lone electron pairs for solvation, where the use of acetonitrile is efficient at disrupting aggregates, which could be an explanation for why we see low MW chromophores absorbing in the blue-UV region.

Considering how molecularly complex BrC is, determining a reference is beyond the scope of our study, but certainly something the community should consider further. However, we hope we have successfully emphasized the importance of understanding and characterizing the factors that can impact the spectral properties of BrC during absorption measurement processes.

The investigation of pH impact on absorption prompted the investigation of other plausible explanations to describe the discrepancies between the online BrC absorption measurements and offline SEC-UV measurements. Major differences between the online and offline analyses are the use of acetonitrile for the chromatographic separation and that the aqueous samples were stored prior to the SEC-UV analysis. The findings highlight that BrC species are more stable collected on filters rather than in aqueous solution and the importance of inter-comparison absorption measurements by multiple methods.

Therefore, in considering possible pH, solvent, and storage effects, section 3.4 was retitled and rewritten:

Section 3.4 was titled:

**Line 287: 3.4 Solvents affect the measured absorption spectra**
This was changed to:

[revised manuscript text omitted]

"Figure S6. Absorption as a function of wavelength measured in various mobile phases of (a) 4-nitrocatechol (b) 4-hydroxy-3-methoxy cinnamaldehyde (c) vanillin (d) 7-hydroxycoumarin and (e) a mixture of the four compounds by a diode array detector. The default mobile phase consists of 1:1 mixture of acetonitrile and DIW with 25 mM ammonium acetate. The purple and black traces represent pH controlled mobile phases, where the pH of the ammonium acetate solution was adjusted to pH 5 and pH 9 prior to the addition of the acetonitrile."

[Figure]

"Figure S14. Size separation of a fresh SRHA solution which was then re-run 20 months later."

Figure 1 displays absorption at 300 nm measured by the SEC-UV for analysis normalized to CO as a function of plume age.  The purpose of the linear fit is to represent whether BrC absorption was increasing or depleting as a function of plume age and to compare these findings to the literature. The linear fit for the flight denoted by the black markers does not extend past the data shown; the data shown in Figure 1 is not cut-off in any way.

3. **For the solvent effects, pH has also been demonstrated to play an important role in shifts in the absorption of BrC. What was the pH (estimated) of the water for the PILS analysis. Do you expect a difference compared to the pH that is found in the water collected for offline analysis? The idea of pH for organics in organic solvents is complicated, but the pH of the water in the online measurements may play a role in the differences observed.**

We agree that pH can play a role in absorption shifting. To investigate the impact the pH of the mobile phase has on wavelength dependent absorption, the ammonium acetate solution was adjusted to pH 5 and pH 9 prior to the addition of the acetonitrile. Several compounds that contribute to brown carbon absorption (4-nitrocatechol, vanillin, 7-hydroxycoumarin, 4-hydroxy-3-methoxy cinnamaldehyde, and mixture of the four compounds) and SRFA were analyzed in the following mobile phases: DIW, DIW with 25 mM ammonium acetate, the default mobile phase (not pH controlled), pH 5, and pH 9. There is red shifting when the pH was greater than the $pK_a$ of the compound. We acknowledge that numerous studies have investigated the impact directly acidifying and alkalinizing a SRFA sample have on absorption – there is a slight increase in absorption with increasing pH (Phillips et al., 2017). These results are shown in new Figure S6.

[Figure]

"**Figure S6. Absorption as a function of wavelength measured in various mobile phases of (a) 4-nitrocatechol (b) 4-hydroxy-3-methoxy cinnamaldehyde (c) vanillin (d) 7-hydroxycoumarin and (e) a mixture of the four compounds by a diode array detector. The default mobile phase consists of 1:1 mixture of acetonitrile and DIW with 25 mM ammonium acetate. The purple and black traces represent pH controlled mobile phases, where the pH of the ammonium acetate solution was adjusted to pH 5 and pH 9 prior to the addition of the acetonitrile."**

The pH of the PILS water was approximately 5. Since the aqueous samples flowed through the liquid waveguide capillary cell prior to its collection in the polypropylene tubes, we do not expect a difference in pH of between the online analysis and aqueous samples. To assess the impact of pH and mobile phase on a complex mixture, we also measured the absorption of a SRFA aqueous solution and a FIREX-AQ aqueous sample with the abovementioned mobile phases (Figure 4 and Figure S8). In contrast to the individual BrC compounds, no major changes in the spectral shape were observed under different mobile phase conditions. To confirm these results, we measured the absorption of SRFA in each solvent using an Agilent spectrophotometer (Figure S7).

[Figure]

**"Figure 4. Absorption as a function of wavelength of (a) SRFA and (b) a FIREX-AQ aqueous sample collected on 28 Aug 2019 L3. The default mobile phase consists of 1:1 mixture of acetonitrile and DIW with 25 mM ammonium acetate. The purple and orange traces represent**

**pH controlled mobile phases, where the pH of the ammonium acetate solution was adjusted to pH 5 and pH 9 prior to the addition of the acetonitrile.”**

[Figure]

**“Figure S7. Absorption as a function of wavelength of SRFA measured using a Agilent 8453 UV-visible Spectroscopy System. A solution of 15 µg/mL SRFA was diluted by 50 % by the mobile phase and then transferred to the cuvette prior to measurement. The default mobile phase consists of 1:1 mixture of acetonitrile and DIW with 25 mM ammonium acetate. The purple and orange traces represent pH controlled mobile phases, where the pH of the ammonium acetate solution was adjusted to pH 5 and pH 9 prior to the addition of the acetonitrile.”**

[Figure]

**“Figure S8. Absorption as a function of wavelength of SRFA measured in a 25 mM ammonium acetate solution and in a mobile phases controlled to pH 5.”**

4. **Solvent effects can play a role in the position of the absorption, but I've seen less discussion around changing the overall intensity of the absorption. Do you think that the increase for the offline measurements can be attributed to this, or are there other possible reasons for that increase?**

The offline SEC-UV analysis and the online absorption measurements by the BrC-PILS were converted to $Mm^{-1}$, a common unit used to express the magnitude of BrC absorption. The total absorption by the SEC-UV measurements is approximately an order of magnitude greater than the BrC-PILS measurements at 300 nm. We are confident that the optical system and the diode array detector can measure absorption that are within agreeance. For example, we compared absorption measurements of 4-nitrocatechol by the LWCC of the BrC-PILS and the diode array detector converted to absorption cross section, showing good good agreement (Figure S13 below).

[Figure]

**"Figure S13. Absorption cross section of 4-nitrocatechol measured by injection onto the LWCC of the BrC-PILS and onto the diode array detector of the SEC-UV set-up compared to Hinrichs et al. 2016."**

The investigation of pH impact on absorption prompted the investigation of other plausible explanations to describe the discrepancies between the online BrC absorption measurements and offline SEC-UV measurements. Major differences between the online and offline analyses are the use of acetonitrile for the chromatographic separation and that the aqueous samples were stored prior to the SEC-UV analysis.

Therefore, in considering possible pH, solvent, and storage effects, section 3.4 was retitled and rewritten:

Section 3.4 was titled: 3.4 Solvents affect the measured absorption spectra

This was changed to:

[revised manuscript text omitted]